# Pd$_x$Ni$_y$/TiO$_2$ Electrocatalysts for Converting Methane to Methanol in An Electrolytic Polymeric Reactor—Fuel Cell Type (PER-FC)

**Jéssica F. Coelho, Isabely M. Gutierrez, Nivaldo G. P. Filho, Priscilla J. Zambiazi** , **Almir O. Neto** and **Rodrigo F. B. de Souza** *

Instituto de Pesquisas Energéticas e Nucleares, IPEN-CNEN/SP, Cidade Universitária, Av. Professor Lineu Prestes, 2242, São Paulo 05508-900, SP, Brazil
* Correspondence: souza.rfb@gmail.com

**Abstract:** Pd$_x$Ni$_y$/TiO$_2$ bimetallic electrocatalysts were used in fuel cell polymeric electrolyte reactors (PER-FC) to convert methane into methanol through the partial oxidation of methane promoted by the activation of water at room temperature. X-ray diffraction measurements showed the presence of Pd and Ni phases and TiO$_2$ anatase phase. TEM images revealed mean particle sizes larger than those reported for PdNi materials supported, indicating that TiO$_2$ promotes particle aggregation on its surface. Information on the surface structure of electrocatalysts obtained by Raman spectra indicated the presence or formation of NiO. The PER-FC tests showed the highest power density for the electrocatalyst with the lowest amount of nickel Pd$_{80}$Ni$_{20}$/TiO$_2$ (0.58 mW cm$^{-2}$). The quantification of methanol through the eluents collected from the reactor showed higher concentrations of methanol produced, revealing that the use of TiO$_2$ as a support also increased the reaction rate.

**Keywords:** methane to methanol; electrocatalysts; polymer electrolyte reactor; fuel cell reactor



## 1. Introduction

Methane, despite being recognized as a primary energy source and a viable solution for the energy transition, also poses a significant challenge due to its status as a greenhouse gas that is more potent than CO$_2$ [1,2]. The greenhouse gas aspect has drawn significant attention from entities and governments; thus, actions to mitigate its emissions have become of great importance. Amongst these strategies, the utilization of methane as a raw material for the production of other molecules has become particularly significant [3,4].

The conversion of methane into various products is a well-known solution that has been in existence for over a century, utilizing the Fischer–Tropsch process, which is industrially applied and has been continually refined over time. However, this process necessitates high temperatures and pressures [5–7], as it is a complex task given methane's low polarizability and high C-H binding energy among hydrocarbons [8–10].

Recently, alternative methods have emerged, such as the utilization of electrochemical reactors incorporating polymeric electrolytes type fuel cell to convert methane into products at moderate temperatures and pressures [4,11]. This approach presents an interesting opportunity as it not only converts methane into products such as methanol but also generates electricity as a byproduct [11,12].

Research conducted by Santos and coworkers [13] has reported that utilizing Pd catalysts enables the conversion of methane into methanol due to PdO's carbophilic properties. Additionally, they found that Ni catalysts were also active, albeit less efficient. Studies [14,15] have also revealed that adding nickel to Pd catalysts can increase methanol production through a synergistic effect of Pd's carbophilic sites and Ni's ability to activate water. This approach also enables the use of cheaper catalysts than palladium.

During partial oxidation of methane, reactive oxygen species (ROS) react with the hydrocarbon. ROS can not only promote hydrocarbon oxidation but also degrade the catalyst support, which is typically made of Vulcan carbon. Thus, replacing it with a support that has high surface area, low cost, and chemical stability, in addition to being able to aid in methanol production, could be beneficial [16]. One such material is anti-mony-tin oxide (ATO), which is known for its stability and good conductivity [17]. ATO has been shown to have promising properties as a catalyst support, particularly for applications involving electrochemical reactions [18]. Another material that has been widely studied for use as a catalyst support is titanium dioxide ($TiO_2$). $TiO_2$ is a widely available, inexpensive, and chemically stable material that has been found to have desirable properties in appropriate proportions [16].

Furthermore, titanium dioxide in anatase form used as support for nickel nanoparticles improves the reducibility of metal particles and increases activity for the production of synthesis gas [19]. The present study investigated the partial oxidation of methane to methanol on PdNi catalysts in a polymeric electrolyte electrochemical reactor, often referred to as a fuel cell.

## 2. Results

The X-ray diffractograms of Pd-Ni supported on $TiO_2$ materials are shown in Figure 1. It is possible to observe in Figure 1a that peaks corresponding to the anatase phase of $TiO_2$ (JCPDS # 21-1272) are present at 2θ values of approximately 25°, 36°, 37°, 38°, 48°, 53°, 55°, 63°, 68°, 70°, 75°, and 78°. However, peaks related to Pd and Ni can only be observed when a logarithm of the intensity is taken, as shown in Figure 1b, due to the large discrepancy in particle sizes. For Pd-containing catalysts, it is possible to identify peaks at 2θ values of approximately 39°, 46°, and 66°, which are associated with the planes (111), (200), and (220) of the Pd face-centered cubic (FCC) structure, according JCPDS # 89-4897. The absence of shifts in the diffraction peaks of PdNi suggests that no alloy formation has occurred. The Ni/$TiO_2$ electrocatalyst exhibits a peak at 2θ of 42°, which is ascribed to NiO (JCPDS # 75-269) and at 2θ values of 44 and 78°, corresponding to Ni (JCPDS # 87-0712). Mateos-Pedrero et al. [17] reported that the presence of a $Ni^0$ phase suggests that $TiO_2$ has the potential to act as a reducing agent for metal particles.

The nanostructure of Pd-Ni supported on $TiO_2$ was analyzed using transmission electron microscopy (TEM), and the histograms of the particle size are shown in Figure 2. The micrograph images revealed an aggregation of nanoparticles on the $TiO_2$ support. However, it was still possible to determine the particle size. The average particle size measured was around 12.4, 5.9, 11.4, and 15 nm, respectively, for $Pd_{80}Ni_{20}$/$TiO_2$, $Pd_{50}Ni_{50}$/$TiO_2$, $Pd_{20}Ni_{80}$/$TiO_2$, and Ni/$TiO_2$. In fact, the Pd and Ni supported in $TiO_2$ and in based nanocatalysts present particle sizes larger than those reported for PdNi materials supported on carbon or ATO ($Sb_2O_5.SnO_2$) [14,15], which suggests that the $TiO_2$ support may promote the aggregation of particles on its surface.

To obtain more information about the material's surface under conditions that closely resemble its intended application, the patterns obtained through the combination of cyclic voltammetry (CV) and in situ Raman spectroscopy can prove useful (as depicted in Figure 3). In n cyclic voltammetry, it is not possible to clearly observe the hydrogen adsorption/desorption region due to the synergistic effect of the metallic oxides present, which can block the surface of palladium [19]. The peak indicating the reduction of metallic oxides at around −0.4 V is observed to shift toward more negative potentials as the amount of Ni in the material increases. This phenomenon can be explained by the stronger adsorption of oxide species on the surface of bimetallic materials in the presence of Ni [20].

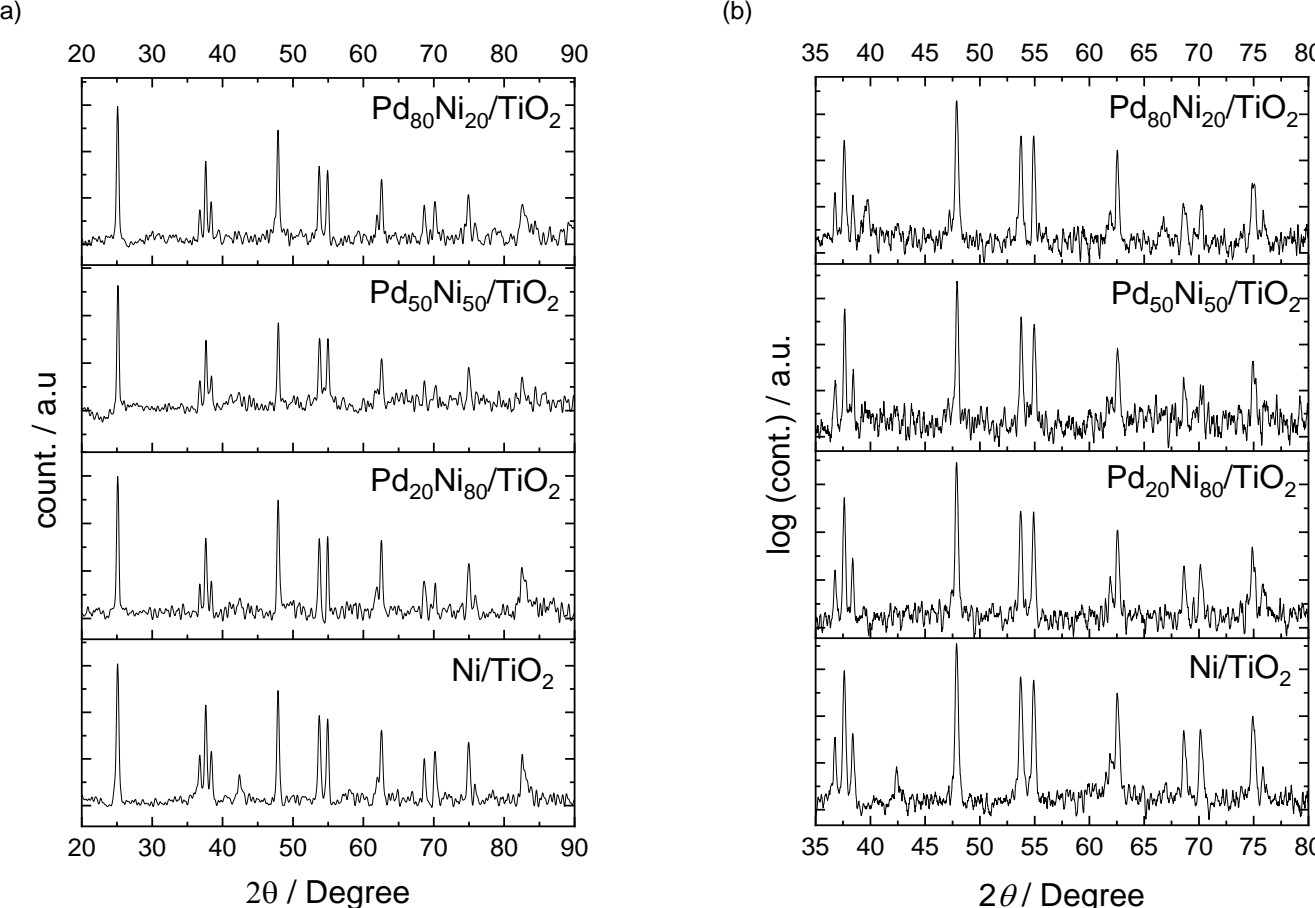

**Figure 1.** (**a**) X-ray diffractogram (XRD) pattern of $Pd_xNi_y/TiO_2$ electrocatalysts; (**b**) logarithm of the intensities of the diffractograms in (**a**).

Figure 3b–f illustrates the in situ Raman spectra obtained during voltammetry, which feature peaks related to the B1g, A1g, and Eg modes of the anatase phase of $TiO_2$ at 397 cm$^{-1}$, 517 cm$^{-1}$, and 638 cm$^{-1}$, respectively [21,22].

In materials containing Pd, the band at 638 cm$^{-1}$ is observed to increase in intensity and width, indicating a convolution of the Eg mode of $TiO_2$ with the 637 cm$^{-1}$ torsion of the PdO-H bond [23]. Additionally, bands with wavelengths of 1314 and 1605 cm$^{-1}$ are present in materials that contain Ni in their composition, indicating the presence or formation of NiO [24], and these bands are observed to increase with potential, with peaks becoming more prominent at less negative potentials.

Figure 4 shows the polarization and power density curves obtained in galvanostatic mode, from the PER-FC reactor. The maximum power density was obtained for $Pd_{80}Ni_{20}/TiO_2$ (0.58 mW cm$^{-2}$), $Ni/TiO_2$ (0.33 mW cm$^{-2}$), $Pd_{20}Ni_{80}/TiO_2$ (0.31 mW cm$^{-2}$), and $Pd_{50}Ni_{80}/TiO_2$ (0.29 mW cm$^{-2}$). The use of $TiO_2$ as a support led to bigger power density compared to the results obtained with electrocatalysts of the same bimetallic compositions of Pd and Ni supported on ATO, prepared in a previous work with Pd-Ni for methane oxidation [15].

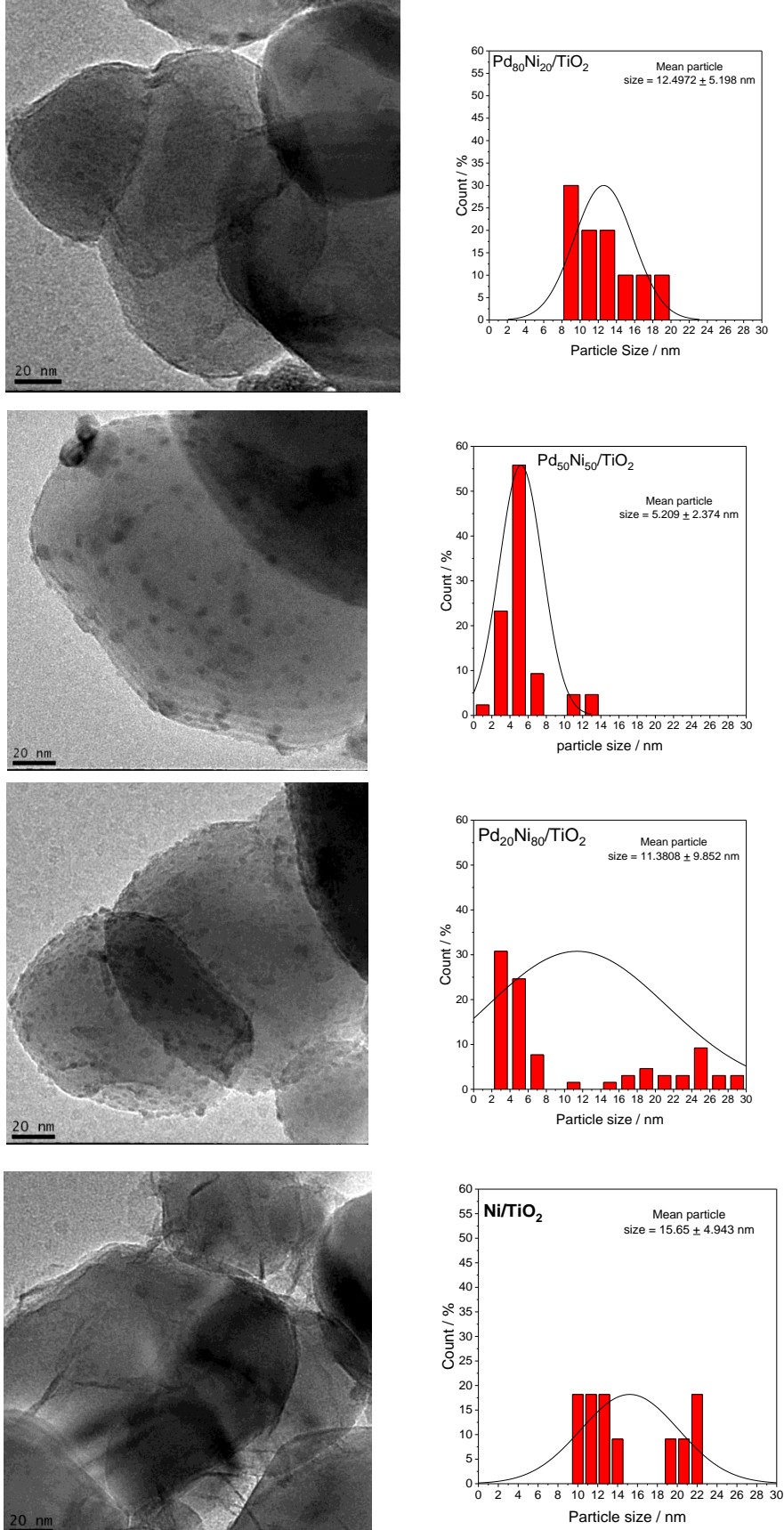

**Figure 2.** TEM Micrograph Images and Particle Size Distribution Histograms for Pd$_x$Ni$_y$/TiO$_2$ electrocatalysts.

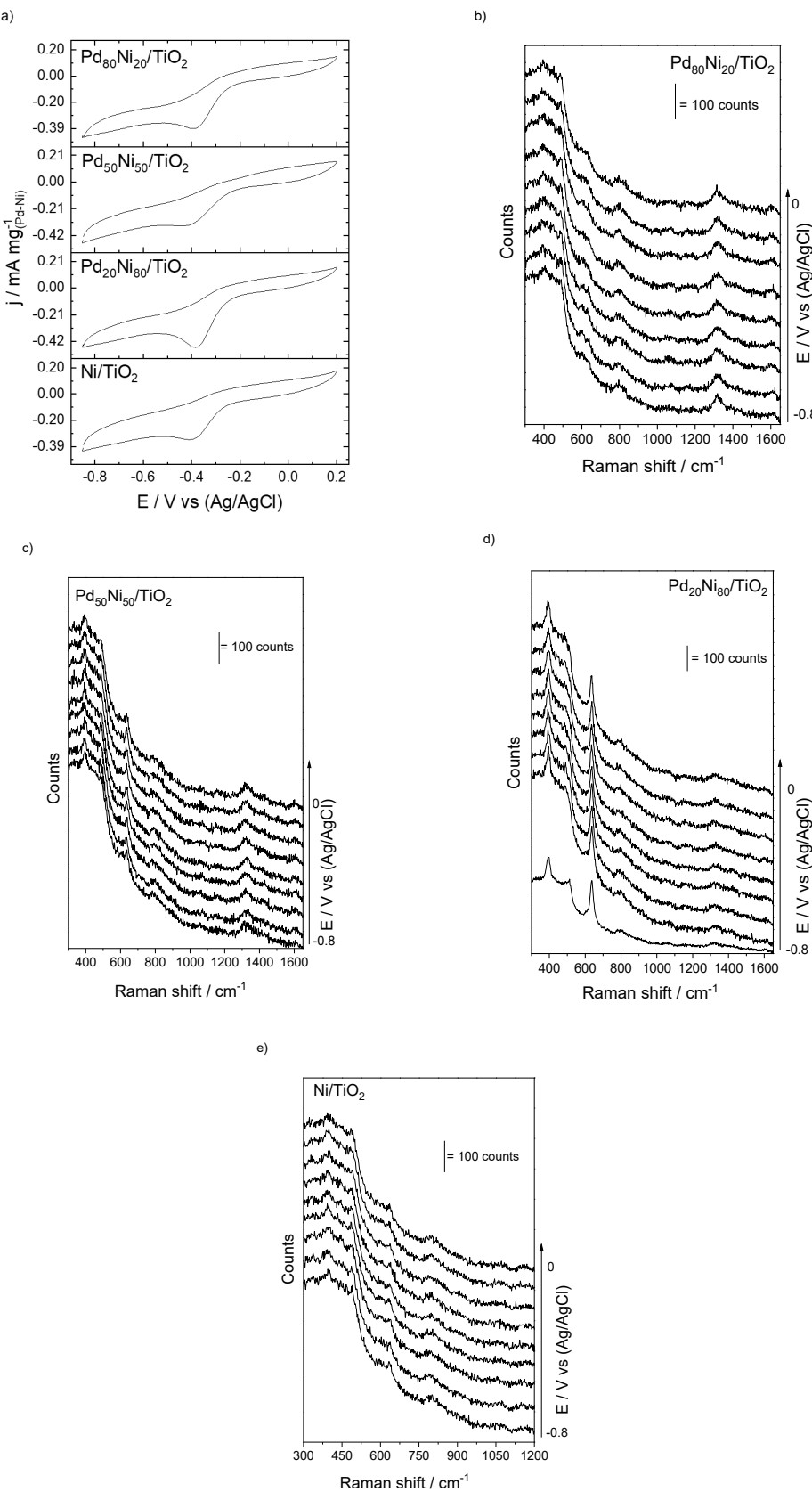

**Figure 3.** (**a**) cyclic voltammetry of $Pd_xNi_y/TiO_2$ electrocatalysts prepared with different metallic proportions in KOH 1 mol $L^{-1}$ with a sweep speed of 10 mV $s^{-1}$ and a potential window of $-8.5$ to 0.2 V. (scan rate v = 10 mV $s^{-1}$) in 1 mol $L^{-1}$ NaOH aqueous solution. (**b**–**e**) The spectrum of in situ Raman-assisted electrochemical measurements collected at different potentials in NaOH 1.0 mol $L^{-1}$.

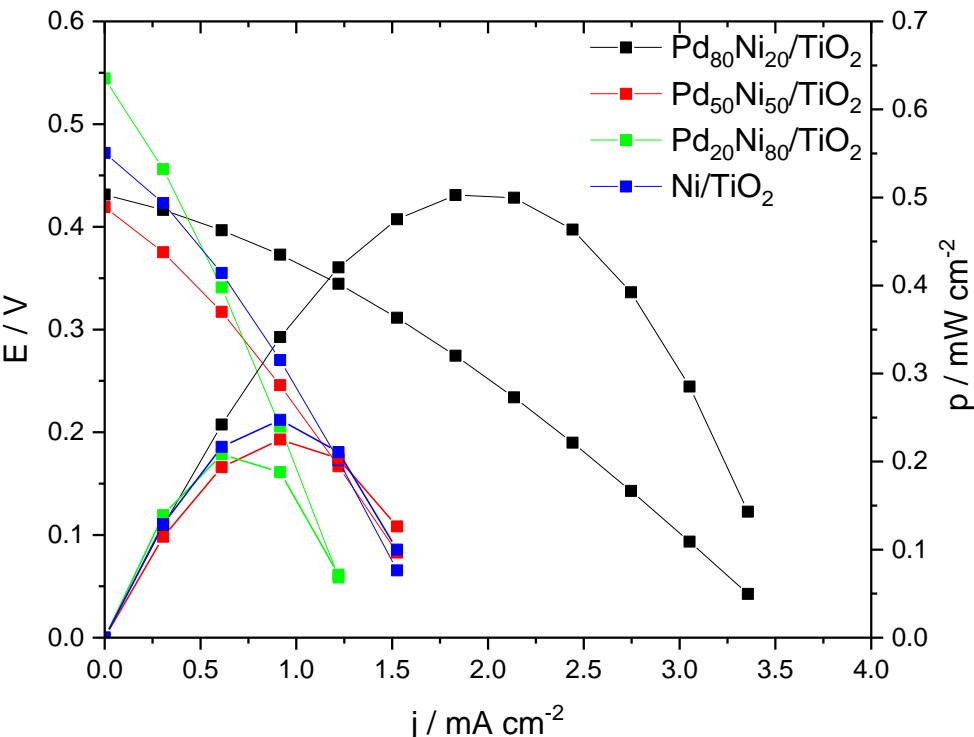

**Figure 4.** Polarization and power density curves for PER-FC composed of anodes containing $Pd_xNi_y/TiO_2$ electrocatalysts with a load of 5 mg cm$^{-2}$; Pt/C Basf cathodes with 1 mg Pt cm$^{-2}$ loading with 20 wt% Pt on carbon, and Nafion 117 membrane treated with KOH 6.0 mol L$^{-1}$. Tests carried out with supply of KOH 1.0 mol L$^{-1}$ + CH$_4$ 50 mL min$^{-1}$ at the anode and O$_2$ with a flow of 200 mL min$^{-1}$ at the cathode.

The samples collected at 100 mV intervals during reactor operation were analyzed utilizing infrared spectroscopy (Figure 5) wherein bands were identified at 1075 and 1030 cm$^{-1}$ relative to methanol [13,25]; it is possible to note that materials with higher amounts of nickel exhibit these bands with less intensity compared to materials with lower proportions of Ni. Materials containing more nickel also present ($Pd_{20}Ni_{80}/TiO_2$ and $Ni/TiO_2$) higher intensities for sodium formate identified by the bands at ~1142 cm$^{-1}$ and 1345 cm$^{-1}$ [26]. The band relating to sodium carbonate (~1375 cm$^{-1}$) [27,28] appears only discreetly in all materials.

Figure 6 shows the HPLC quantification of methanol obtained from effluents collected from the PER-FC reactor by applying potentials of 100 mV from the OCP to 0 V, using $Pd_xNi_y/TiO_2$ bimetallic electrocatalysts in different metallic proportions for methane oxidation. As can be seen, the reaction rate, obtained by Equation (1), points to a decrease in methanol production with a decrease in the proportion of palladium and an increase in the proportion of nickel.

$$r = \frac{\text{Methanol}_{\text{amount}}}{\text{Volume} \times \text{Time}} \tag{1}$$

Compared to other materials reported in the literature, the use of TiO$_2$ as a catalyst support appears to enhance the activity of supported materials. For example, in this study, $Pd_{80}Ni_{20}/TiO_2$ demonstrated a significantly higher rate reaction of 14.5 mol L$^{-1}$ h$^{-1}$ compared to $Pd_{70}Ni_{30}/C$, which had a rate reaction of 3.5 mol L$^{-1}$ h$^{-1}$, $Pd_{90}Ni_{10}/C$, which had a rate reaction of approximately 3 mol L$^{-1}$ h$^{-1}$ [14], and $Pd_{80}Ni_{20}/ATO$, which had a rate reaction of 6.5 mol L$^{-1}$ h$^{-1}$ [15]. The enhanced activity of $Pd_xNi_y/TiO_2$ is probably due to the synergistic effect of the carbophilic sites of Pd with the generation of reactive oxygen species (ROS). However, it should be noted that materials with the highest proportions of nickel became less active or even inactive, possibly due to the properties of TiO$_2$ in water

activation, creating an imbalance between the types of sites and causing steric hindrance for new water molecules to be activated.

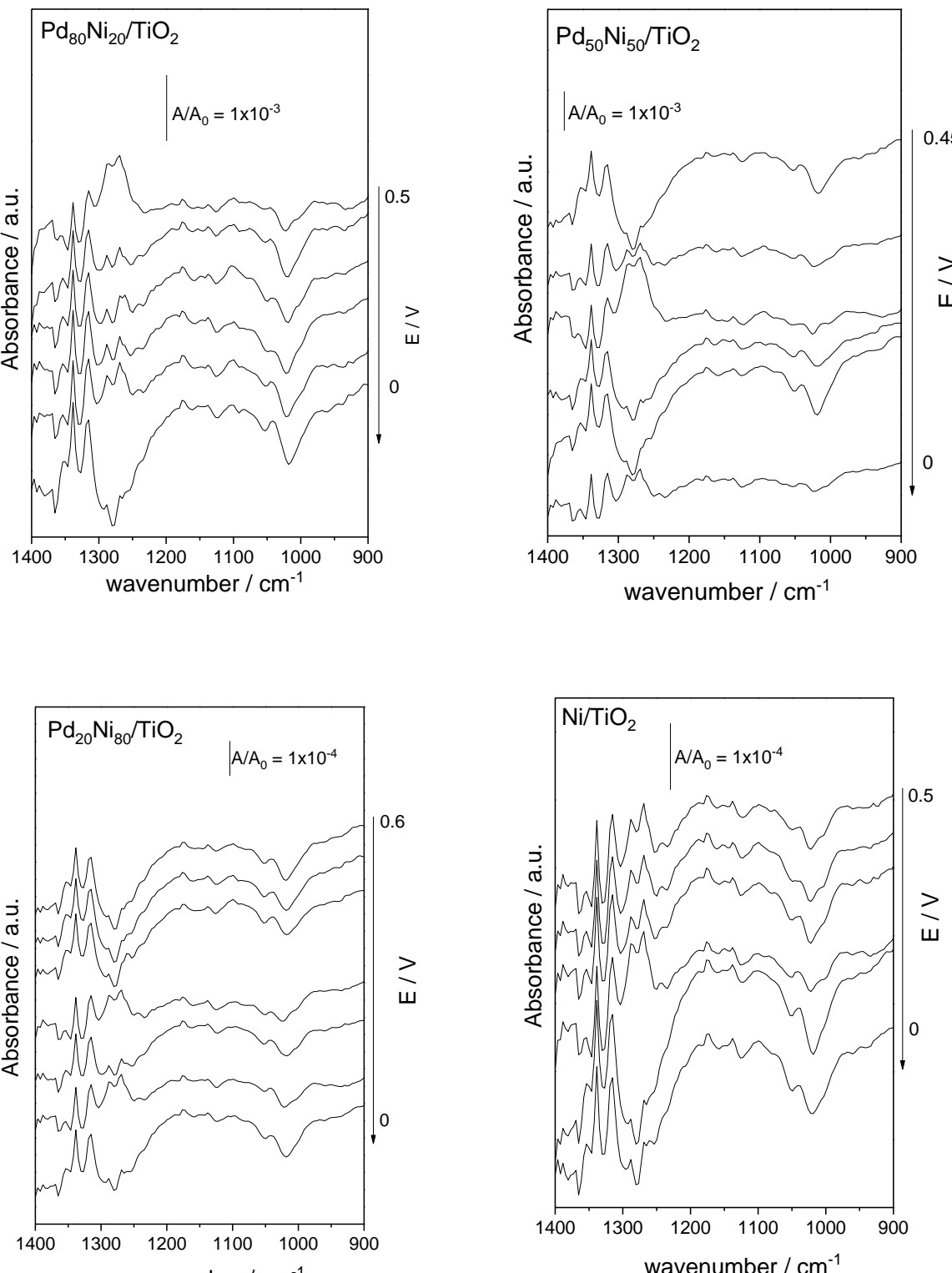

**Figure 5.** FTIR spectra of PER-FC effluents with $Pd_xNi_y/TiO_2$ anode collected for 5 min for each application of potentials of $-0.5$; $-0.4$; $-0.3$; $-0.2$; $-0.1$ and 0 V.

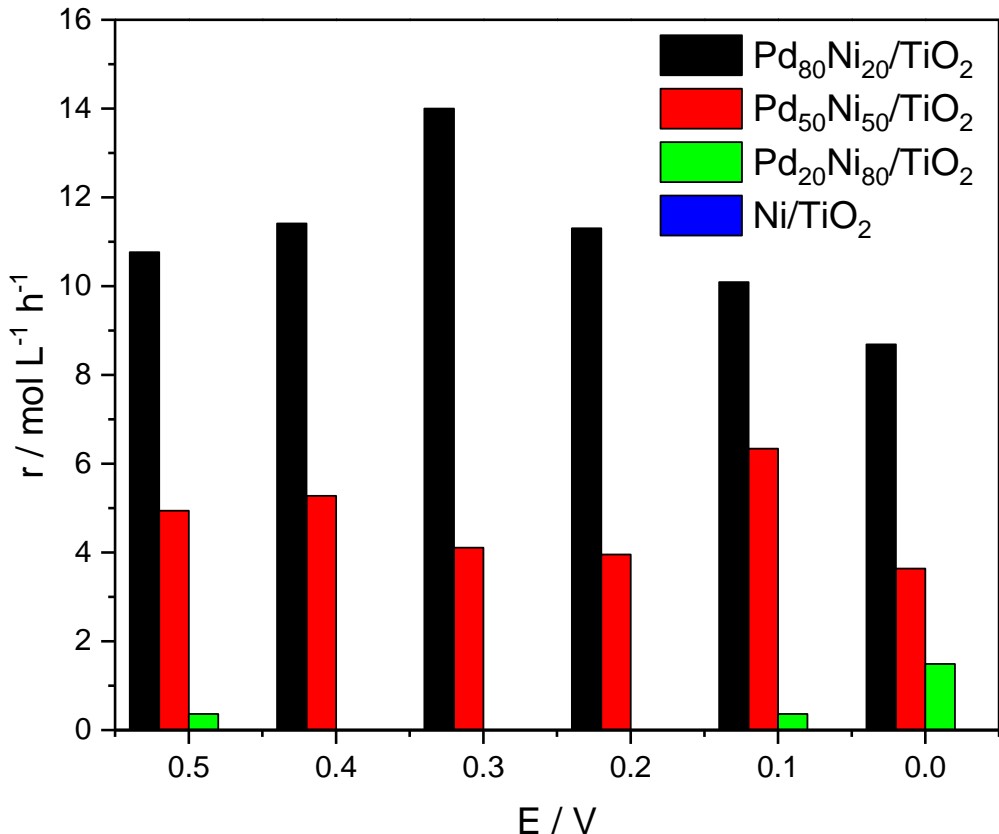

**Figure 6.** Reaction rates (in mol $L^{-1}$ $h^{-1}$) for methanol production in $Pd_xNi_y/TiO_2$ anodic electrocatalysts at different potentials.

### 3. Materials and Methods

The preparation of Pd-Ni supported on $TiO_2$ (20% *w/w*) catalysts of varying compositions was achieved via the sodium borohydride reduction method [15], utilizing a mixture of ultrapure water, 1:1 isopropanol, and Aldrich $TiO_2$ anatase, which was stirred throughout the process. The metallic precursors, $Pd(NO_3)_2.2H_2O$ (Aldrich) and $NiCl_2.6H_2O$ (Aldrich), were added to the mixture in the appropriate amounts. Subsequently, an aqueous solution containing $NaBH_4$ (Aldrich) in an excess of 5:1 relative to the metals present in the mixture was added, and the stirring was maintained for 30 min. Following this, the material obtained was filtered and thoroughly washed with ultrapure water.

The materials were physically characterized by X-ray diffraction (XRD) using a Rigaku—Minifex II diffractometer with a Cukα radiation source of 0.15406 Å; the analysis conditions were defined in the range of 20° to 90°, with a scan speed of 2° $min^{-1}$. Transmission electron microscopy (TEM) performed by a JEOL JEM-2100 electron microscope operated at 200 keV, and the histograms were made with the measurement of 300 nanoparticles of each catalyst.

The electrocatalysts were characterized using an in situ electrochemical technique assisted by Raman spectroscopy. The equipment employed for this purpose included a potentiostat Autolab PGstat 302N and a Raman Macroram-Horiba spectrometer with a laser of 785 nm. These experiments were conducted within a three-electrode electrochemical cell setup [29], which consisted of a working electrode constructed of 0.2 $cm^2$ diameter glassy carbon, covered with an ultra-fine porous layer produced from an ink that was prepared during each experiment. The ink was composed of 5 mg of the synthesized $Pd_xNi_y/TiO_2$ electrocatalysts, 600 μL of ultrapure water, 900 μL of isopropanol, and 25 μL of Nafion® (D-520), which were mixed in an ultrasound bath. The Ag/AgCl (3 mol $L^{-1}$) was used as a reference electrode and a counter electrode of platinum (2 $cm^2$).

Assays for the conversion of methane to methanol were carried out in a fuel cell polymer electrolyte reactor (PEF-FC) featuring a membrane electrode assembly (MEA) with 5 cm$^2$ electrodes, constructed using 1 mg/cm$^2$ of Pd$_x$Ni$_y$/TiO$_2$ at the anode, a Nafion 117 membrane treated with 6.0 mol/L NaOH as the electrolyte, and 1 mg/cm$^2$ of Pt/C Basf (20% $w/w$) as the cathode. The electrodes were prepared by depositing electrocatalytic ink on carbon cloth treated with PTFE. For each experiment, 25 mg of electrocatalyst was utilized for the anode and cathode, respectively, and 292 µL of a Nafion D-520 (Aldrich) solution was mixed in ultrasound and applied on carbon cloth by brushing. The reactor was fed simultaneously with 50 mL/min of methane and 1 mL/min of KOH at room temperature at the anode, while the cathode was fed with O$_2$ humidified with water at 85 °C with a flow rate of 200 mL/min. Effluents from the reactor were collected in aliquots every 100 mV for 120 s from the open circuit potential (OCP) to 0 V and analyzed using high performance liquid chromatography (HPLC) on a YoungIn Chromass YL9100 (HPLC) with UV/Vis detector, with detection carried out at 205 nm. Chromatography experiments were conducted using a flow of 0.8 mL/min of 50% water and 50% acetonitrile in an isocratic run on a C18 column (Phenomenex Luna 5 µm, 250 × 4.6 mm). The calibration curve follows the equation area = 59.916 + 238.59 [methanol], with r$^2$ = 0.9981. The samples were characterized using infrared spectroscopy (FTIR) performed on a Nicolett$^®$ 6700 with an ATR Miracle (Pike) accessory and diamond/ZnSe crystal and an MCT detector.

## 4. Conclusions

The utilization of TiO$_2$ as a catalyst support presents itself as a viable alternative to carbon, owing to its chemical and electrochemical stability in the partial oxidation reaction of methane to methanol. The X-ray diffractograms of Pd-Ni supported on TiO$_2$ materials showed peaks corresponding to the anatase phase of TiO$_2$, and peaks related to Pd and Ni were identified. The nanostructure of Pd-Ni supported on TiO$_2$ was analyzed using TEM, and the particle size was determined to be around 12 nm for different compositions. The products collected during reactor operation were analyzed using infrared spectroscopy, and materials with higher amounts of nickel exhibited bands with less intensity compared to materials with lower proportions of Ni; the better methanol rate reaction obtained was about 14 mol L$^{-1}$ h$^{-1}$ over Pd$_{80}$Ni$_{20}$/TiO$_2$. In the Pd-Ni-TiO$_2$ system, the optimal composition for the production of methanol is more intricate, and it has been determined that Ni must be incorporated only as a dopant to produce a highly efficient catalytic material. Furthermore, it has been observed that an excessive amount of Ni decreases the catalytic activity of the material.

**Author Contributions:** J.F.C., synthesis of electrocatalysts, measurements in a polymeric electrolyte reactor, DRX analysis, development of methodology for HPLC, analysis of products by HPLC, FTIR; I.M.G., measurements voltammetric and Raman in situ; N.G.P.F., measurements voltammetric and Raman in situ; P.J.Z., planning, discussing, and writing the draft; A.O.N., oversight and leadership responsibility for the research activity planning and execution; R.F.B.d.S., management and coordination for the research activity planning and writing of the final draft. All authors have read and agreed to the published version of the manuscript.

**Funding:** This research was funded by CAPES, CNPq (302709/2020-7), FAPESP (2017/11937-4) and CINE-SHELL (ANP).

**Informed Consent Statement:** Not applicable.

**Data Availability Statement:** Data can be made available if requested directly from the authors.

**Conflicts of Interest:** The authors declare no conflict of interest.

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
