# Peer review of "PdxNiy/TiO2 Electrocatalysts for Converting Methane to Methanol in An Electrolytic Polymeric Reactor—Fuel Cell Type (PER-FC)"

_methane, doi:10.3390/methane2020011_

Round 1
Reviewer 1 Report
Dear Authors,
I have reviewed the manuscript and recommend accepting the publication in its present form. The manuscript presents a well-written and well-structured analysis using PdxNiy/TiO2 to convert methane into methanol through the partial oxidation of methane promoted by water activation at room temperature.
The study design is appropriate, and the data collection and analysis are thorough and well-explained. The authors have provided clear and concise results and conclusions, which are well-supported by the data presented. The manuscript also includes an insightful discussion highlighting the quantification of methanol through the eluents collected from the reactor showed significant production. Moreover, the use of TiO2 as support increases the kinetic.
The manuscript is well-organized, and the language is clear and easy to understand. The references cited are appropriate and up-to-date and follow the journal's scope.
Overall, the manuscript contributes to a wide challenge: converting methane into methanol.
However, the authors may have the quantification of the conversion of methane in methanol and do conclusion improving.
Therefore, I recommend accepting the publication in its present form.
Author Response
“However, the authors may have the quantification of the conversion of methane in methanol and do conclusion improving.”
A: The rate reactions for different compositions were calculated based on methanol quantification data. However, reporting only production values for a flow-operating system did not make sense. Therefore, we chose to report the quantifications in the form of rate reactions, resulting in an improved conclusion.
Reviewer 2 Report
· Please highlight the objective of this study in the abstract.
· The introduction part doesn’t provide sufficient information about the previous work. In addition, the novelty of this work isn’t well clarified.
· Please add the JCPDS of Pd, Ni, and NiO in Figure 1 to make it clear regarding the absence/presence of their peaks.
· In the title, the authors wrote PdxNiy/TiO2 electrocatalyst which means an alloy is formed between Ni and Pd. According to the XRD results, the authors claimed that no alloy is formed between Ni and Pd (page 2, lines 66 and 67). Please, clarify this point.
· Why adding Pd to Ni/TiO2 resulted in decreasing the average particle size in Pd20Ni80/TiO2 and Pd50Ni50/TiO2?
· In Figure 2, please distinguish the supported nanoparticles (NPs) of the active phase from the support. In TEM images of Pd80Ni20/TiO2 and Ni/TiO2, it is very hard to see the supported NPs for which the authors calculated the particle size.
· The highest power density was obtained over the catalyst, which have the highest content of Pd (Pd80Ni20/TiO2), which isn’t beneficial for the large-a scale production because Pd is a noble metal and hence the production cost will be high. In addition, it is expected that the higher the content of the noble metal, the higher the activity. Then, what is the novelty here?
· Since the authors want to clarify the role of using TiO2 as a support, it is recommended to compare the activity of the best-performing catalyst in this study (Pd80Ni20/TiO2) when another support is used, e.g., carbon (use another support for Pd80Ni20 instead of TiO2). This will give a clear idea of whether TiO2 is beneficial as catalyst support for Pd80Ni20 or not.
Author Response
- The introduction part doesn’t provide sufficient information about the previous work. In addition, the novelty of this work isn’t well clarified.
A: introduction was improve
- Please add the JCPDS of Pd, Ni, and NiO in Figure 1 to make it clear regarding the absence/presence of their peaks.
A: We opted not to include the JCPDS in the original version because the proximity of the peaks would result in a cluttered visualization, requiring the creation of additional frames for legibility. If there were only a few peaks with greater distances, we would have included them already.
- In the title, the authors wrote PdxNiy/TiO2 electrocatalyst which means an alloy is formed between Ni and Pd. According to the XRD results, the authors claimed that no alloy is formed between Ni and Pd (page 2, lines 66 and 67). Please, clarify this point.
A: This nomenclature is also used for binary materials, mainly in the area of electrocatalysis, these binary materials can be alloys, composites, decorated, and the index indicates the proportion of one metal in relation to the other.
- Why adding Pd to Ni/TiO2 resulted in decreasing the average particle size in Pd20Ni80/TiO2 and Pd50Ni50/TiO2?
A: When synthesizing nanoparticles using a combination of a noble metal and a less noble metal, it is typically observed that the average nano-particle size increases. However, in the specific case of a Pd-Ni-TiO2 system, the particle size is larger when high proportions of Pd are present, as compared to when there are smaller proportions of Pd in Ni. Conversely, when there is significantly more Ni than Pd, the particle size increases once again. This phenomenon can be explained by the relative thermodynamic stability of the metals and their ability to form alloys and composites. Pd is a more noble metal than Ni, and therefore possesses a higher thermodynamic stability. When Pd is present in high proportions with TiO2, it is able to stabilize the particles and prevent them from growing excessively. This observation leads to the possibility of identifying an optimal composition of Pd-Ni on TiO2 that could yield a minimum particle size. However, when there is more Ni than Pd, the Pd is not able to stabilize the particles as effectively, leading to an increase in particle size. These parameters have been discussed in the literature. For exemple in N.T.K. Thanh, N. Maclean, S. Mahiddine, Mechanisms of Nucleation and Growth of Nanoparticles in Solution, Chemical Reviews 114(15) (2014) 7610-7630. 10.1021/cr400544s.
.
- In Figure 2, please distinguish the supported nanoparticles (NPs) of the active phase from the support. In TEM images of Pd80Ni20/TiO2 and Ni/TiO2, it is very hard to see the supported NPs for which the authors calculated the particle size.
A To estimate the average size for statistical purposes, more than 300 particles were measured. For this purpose, a large number of TEM images were used, and the most representative images of the material were chosen. However, both Pd80Ni20/TiO2 and Ni/TiO2 materials exhibit significant agglomeration, resulting in a limited number of NPs outside of these regions. This is unfortunate, and it may be attributed to the effects mentioned in the previous response.
- The highest power density was obtained over the catalyst, which have the highest content of Pd (Pd80Ni20/TiO2), which isn’t beneficial for the large-a scale production because Pd is a noble metal and hence the production cost will be high. In addition, it is expected that the higher the content of the noble metal, the higher the activity. Then, what is the novelty here?
A: The main objective of the work is not to achieve high power density, and in the case of pure Pd, even when it does obtain higher power density, it is usually associated with the consumption of the produced molecules. The power density curves are important to provide the reader with a broader perspective on the amount of energy produced as a byproduct of the partial oxidation reaction of methane to methanol.
Currently, we know that palladium is approximately 2000 times more expensive than nickel, so each milligram of palladium is important in the material cost composition. However, the pricing relationships of metals over time are not a good parameter as it depends on the market, but it should be taken into consideration.
Another point that cannot be denied is the relative abundance of nickel, but its durability as a catalyst is inferior to palladium. As a scientific article, the manuscript presents data on binary materials where it may be economically interesting to use a Pd20Ni80/TiO2 catalyst, which, despite its low kinetic factor, is relatively inexpensive for mass production. However, we know that scientific literature does not dictate the market but accumulates data that can guide market developments and/or serve as a database to explain similar effects in other reactions that bear some resemblance to the one exposed here.
- Since the authors want to clarify the role of using TiO2 as a support, it is recommended to compare the activity of the best-performing catalyst in this study (Pd80Ni20/TiO2) when another support is used, e.g., carbon (use another support for Pd80Ni20 instead of TiO2). This will give a clear idea of whether TiO2 is beneficial as catalyst support for Pd80Ni20 or not.
A: A direct comparison can be made between the results presented in this manuscript and the article "M.C.L. Santos, C.M. Godoi, H.S. Kang, R.F.B. de Souza, A.S. Ramos, E. Antolini, A.O. Neto, Effect of Ni content in PdNi/C anode catalysts on power and methanol co-generation in alkaline direct methane fuel cell type, Journal of Colloid and Interface Science 578 (2020) 390-401.https://doi.org/10.1016/j.jcis.2020.06.017", which is the reference 14 of the current manuscript, and studied PdNi/C catalysts under conditions very similar to those presented here. It is worth noting that when supported on TiO2, the activity of partial oxidation of methane to products increased 4 times compared to when supported on carbon, but the power density was much lower, as mentioned in the manuscript.
Reviewer 3 Report
The manuscript «PdxNiy/TiO2 electrocatalysts for converting methane to methanol in an electrolytic polymeric reactor – fuel cell type (PER-FC)» represents study the partial oxidation of methane to methanol on PdNi catalysts in a polymeric electrolyte electrochemical, authors compered catalysts with different Pd/Ni ratio. Although, it is an interesting study, there are several aspects that must be reviewed before it can be accepted for publication.
Figure 1 should be improved. Reflections of TiO2, Pd and Ni should be marked.
In conclusions «The utilization of TiO2 as a catalyst support presents itself as a viable alternative to carbon, owing to its chemical and electrochemical stability in the partial oxidation reaction of methane to methanol.» However authors do not catalytic results between catalysts supported on TiO2 and carbon in the manuscript
Typo lineы 167-167 «…even inactive. possibly due»
Author Response
- Figure 1 should be improved. Reflections of TiO2, Pd and Ni should be marked.
A: We opted not to include the JCPDS in the original version because the proximity of the peaks would result in a cluttered visualization, requiring the creation of additional frames for legibility. If there were only a few peaks with greater distances, we would have included them already.
- In conclusions: “The utilization of TiO2 as a catalyst support presents itself as a viable alternative to carbon, owing to its chemical and electrochemical stability in the partial oxidation reaction of methane to methanol.” However authors do not catalytic results between catalysts supported on TiO2 and carbon in the manuscript
Typo lineы 167-167 «…even inactive. possibly due»
A: The addition of TiO2 to the catalyst increases by up to 4 times the catalytic activity of partial oxidation of methane to methanol in similar compositions with high contents of Pd (70% to 90%). However, in the presence of large amounts of Ni, the catalyst starts to promote total oxidations of the products, and since it occurs via a non-faradaic pathway, it does not reflect in higher power density. The discussion paragraph of the results has been rewritten to better clarify this occurrence.
Reviewer 4 Report
This manuscript reported a PdxNiy/TiO2 bimetallic electrocatalysts and uesd it in fuel cell polymeric electrolyte reactors to convert methane into methanol through the partial oxidation of methane. The author revealed the synergistic effect for Ni and Pd in the optimal composition of PdxNiy/TiO2 through in situ Raman. It was interesting to find that Ni must be incorporated only as a dopant to produce a highly efficient catalytic material while an excessive amount of Ni decreases the catalytic activity of the material in the partial oxidation reaction of methane to methanol. This work focused on the synergistic effect of bimetallic electrocatalysts, the catalysts showed a good performance and the authors provided detailed characterization analysis and solid control experiments for supporting catalytic results. Therefore, I recommend acceptance, subject to the following comments and suggestions by minor revision:
1. Why TiO2 was selected as the support for bimetallic catalysts in electrocatalytic reactions? As we know, TiO2 had not a good electrical conductivity. In addition to this, the authors also found that TiO2 promotes particle aggregation on its surface and did not benefit for exposing the active catalytic sites.
2. The ICP (Inductive Coupled Plasma Emission Spectrometer) data were very important to prove the accurate composition of PdxNiy/TiO2 materials, it was suggested to add ICP data in the main text.
3. In Figure 6, the Pd20Ni80/TiO2 catalyst showed catalytic performance at 0.5, 0.1 and 0.0 V while did not show catalytic performance at 0.2-0.4 V, it was a strange trend, please explain it.
4. It would be better to provide the table of performance comparison with the reported catalysts.
5. The minor error, the full name of ATO in the abstract should be given.
6. It is impressive the synergistic effect of bimetallic electrocatalysts and TiO2 support. The effect of support might be different in other supported catalysts. The recent typical works might help authors understand the effect. (“One-Pot 3D Printing Robust Self-Supporting MnOx/Cu-SSZ-13 Zeolite Monolithic Catalysts for NH3-SCR”, DOI: 10.31635/ccschem.021.202100942; “Co nanoparticles/N-doped carbon nanotubes: Facile synthesis by taking Co-based complexes as precursors and electrocatalysis on oxygen reduction reaction”, DOI: 10.1016/j.colsurfa.2022.129912). If the authors are interesting in the electrocatalytic conversion of high-value chemicals, this relevant work may be helpful (“Heterogeneous nanocomposites consisting of Pt3Co alloy particles and CoP2 nanorods towards high-efficiency methanol electro-oxidation”, DOI: 10.1002/smm2.1032).
Author Response
- Why TiO2 was selected as the support for bimetallic catalysts in electrocatalytic reactions? As we know, TiO2 had not a good electrical conductivity. In addition to this, the authors also found that TiO2 promotes particle aggregation on its surface and did not benefit for exposing the active catalytic sites.
A: TiO2 is not a good conductor and tends to agglomerate, but it possesses a high surface area and is more resistant to corrosion by reactive oxygen species. In studies where it was used as a support for Pt for alcohol oxidation, it showed the formation of more oxygenated products, indicating a propensity for water activation. Given that the spontaneous partial oxidation of methane under mild conditions heavily relies on water activation for the formation of reactive oxygen species, TiO2 has become an excellent option for obtaining products rather than electrical energy, which is merely a byproduct.
- The ICP (Inductive Coupled Plasma Emission Spectrometer) data were very important to prove the accurate composition of PdxNiy/TiO2 materials, it was suggested to add ICP data in the main text.
A: The NaBH4 reduction method is highly reproducible and does not cause selective loss of one metal over another. However, prior to conducting further experiments, the composition is confirmed using EDS, which, although less sensitive than ICP, can measure the relative proportion between metals with great accuracy. If the results fall within the expected parameters, the study with these materials continues, and thus the material corresponds to the reported composition.
- In Figure 6, the Pd20Ni80/TiO2 catalyst showed catalytic performance at 0.5, 0.1 and 0.0 V while did not show catalytic performance at 0.2-0.4 V, it was a strange trend, please explain it.
A: At these potentials, methanol amounts were found to be below the detection limit of the obtained analytical curve, which is likely related to methanol consumption inside the reactor, whether through Faradaic oxidation or not. This is confirmed by the variation in the carbonate bands observed in Figure 5.
- It would be better to provide the table of performance comparison with the reported catalysts.
A: While providing a table of performance comparison with the reported catalysts could be helpful, it is not always necessary, and peripheral information that could be of interest to other researchers may be lost. Therefore, we chose to include the data in the results discussion. This approach improved the discussion by providing comparison values and commenting on them within the text. For data that are not part of the manuscript's results, graphs are a richer source of information because they show the evolution of the conditions leading up to that point in addition to the raw data.
- The minor error, the full name of ATO in the abstract should be given.
A: was corrected
- It is impressive the synergistic effect of bimetallic electrocatalysts and TiO2 The effect of support might be different in other supported catalysts. The recent typical works might help authors understand the effect. (“One-Pot 3D Printing Robust Self-Supporting MnOx/Cu-SSZ-13 Zeolite Monolithic Catalysts for NH3-SCR”, DOI: 10.31635/ccschem.021.202100942; “Co nanoparticles/N-doped carbon nanotubes: Facile synthesis by taking Co-based complexes as precursors and electrocatalysis on oxygen reduction reaction”, DOI: 10.1016/j.colsurfa.2022.129912). If the authors are interesting in the electrocatalytic conversion of high-value chemicals, this relevant work may be helpful (“Heterogeneous nanocomposites consisting of Pt3Co alloy particles and CoP2 nanorods towards high-efficiency methanol electro-oxidation”, DOI: 10.1002/smm2.1032).
A: These are interesting studies, and much of what is reported in them can and will be applied in future work. Thank you for the suggestion.
Round 2
Reviewer 2 Report
The authors made some changes to the manuscript and provided clear answers to my comments. I accept the manuscript in its current form for publication in Methane journal.
Best regards,